# Insights into Putative Health Implications of Gelam (*Melaleuca cajuputi*) Honey: Evidence from *In-Vivo* and *In-Vitro* Studies

**DOI:** 10.3390/medsci4010003

**Published:** 2016-02-23

**Authors:** Boon Keng Chan, Hasnah Haron

**Affiliations:** Nutritional Sciences Programme, School of Healthcare, Faculty of Health Sciences, Universiti Kebangsaan Malaysia, Jalan Raja Muda Abdul Aziz, Kuala Lumpur 50300, Malaysia; nicholas1015@yahoo.com

**Keywords:** Gelam honey, anti-oxidative, antimicrobial, anti-inflammatory, anti-diabetic, antitumor, wound healing, physiochemical, compositional

## Abstract

Honey has been used as a therapeutic agent since ancient times for health maintenance and the treatment of various ailments. In modern days, researchers reappraised the therapeutic values of honey, such as anti-oxidative, anti-inflammatory, antimicrobial, anti-diabetic, anti-tumor, and wound healing properties. These findings supported its applications in the modern healthcare system as complementary medicine. Gelam honey (GH) is a monofloral Malaysian honey which has been proven to have considerable health benefits. This paper presents a state of the art review on the therapeutic values of GH. A descriptive elucidation is performed to elaborate a wide spectrum of biological activities of GH using evidence from a considerable body of literature. The compositional and physiochemical characteristics of GH have contributed substantially to its putative biological properties. A brief explanation will be presented on GH attributes to familiarize readers with this novel natural health product.

## 1. Introduction

Honey is a natural concentrated sugar mixture with a series of medicinal properties, such as anti-inflammatory [1], wound healing [2], anti-oxidative [3,4], anti-diabetic [5,6], antimicrobial activities [7], and antitumor properties [8,9]. Honey has been used since ancient times, as evidenced by apitherapy as one of the branches in folk medicine for the promotion and maintenance of health as well as treatments of ailments [10]. Honey from different botanical and geographical origins may lead to physicochemical and compositional differences, and ultimately affect their biological activities. Gelam honey (GH) is a Malaysian honey produced by *Apis dorsata* bees and its main nectar and pollen source is from the flowers of the *Melaleuca cajuputi* plant, locally known as the Gelam tree. Pollen analysis of GH revealed the primary predominant pollen species was *M. cajuputi,* which accounts for about 54% of total pollen frequency [11]. Being a wild monofloral honey, GH is produced widely in the state of Terengganu on the eastern coast of peninsular Malaysia where Gelam trees are abundant [12]. Recently, GH has received substantial attention from researchers. A body of research has been conducted to characterize the putative health benefits of GH in treatments of various ailments. Within these two decades, GH has come to an exciting stage in healthcare research, in conjunction with the discovery of novel natural therapeutic agents. The present work aims to provide an up-to-date picture of GH utilization in medical research. Comprehensive literature has been summarized and outlined to give a better elucidation in order to familiarize readers with pertinent evidence related to novel health implications of GH.

## 2. Physicochemical Properties of GH

Physicochemical attributes of honey are considered to be of great importance, as they reflect the quality of honey. Table 1 shows the physicochemical characteristics of GH based on previous literature [12,13,14,15]. The pH value of GH was similar to other Malaysian honey and was more acidic than Manuka honey from New Zealand [12]. The acidity of honey results from the fermentation of sugar into organic acid, which contributes to honey’s flavor and stability against microbial spoilage [16]. The moisture content (≤20%) and hydroxymethylfurfural (HMF) content (≤80 mg/kg) were within the limit prescribed by the international regulations for honey [17]. Some studies reported moisture contents slightly higher than the permitted range [14,15], which might be ascribed to the harvest season of the honey. Concerning this subject, gamma irradiation was reported as an effective method to reduce moisture content, thus preserving the freshness of honey through reduction of fermentation rate [15].

## 3. Chemical Composition of GH

Honey has a complex chemical composition which consists of more than 200 substances, including sugars, amino acids, vitamins, minerals, polyphenols, and enzymes [18]. Numerous studies have been conducted to quantify the nutritional composition as well as some non-nutritive constituents, such as enzymes, in GH [12,13,19,20,21]. Table 2 shows the summarized chemical composition of GH.

## 4. Antioxidant Activities of GH

Free radicals have been well documented to have detrimental implications on human health, resulting in a number of pathological conditions such as cancer [24], cardiovascular disease [25], impaired wound healing [26], and gastrointestinal inflammatory diseases [27]. Antioxidants present in natural foods such as honey are recommended in order to scavenge the free radicals. A battery of antioxidant assays is employed to evaluate the antioxidant capacity of a food in the laboratory. However, most studies commonly used DPPH free radical-scavenging activity and ferric reducing antioxidant power (FRAP) assay to determine the antioxidant capacity of GH. A comparative study was conducted to evaluate the antioxidant capacities of GH and coconut honey [28]. The results showed that GH has a free radical scavenging ability of 36.7 × 10^−4^ microequivalent (µeq) at a 4 mg/mL honey extract concentration, where one microequivalent is the ability to reduce one micromole (µM) of pro-oxidants. Meanwhile, the total antioxidant power of GH assessed using the FRAP assay was 13.45 × 10^2^ µM Fe (II) at the honey extract concentration of 4 mg/ml. The antioxidant capacity of GH was significantly (*p* < 0.001) higher than that of coconut honey. The difference might be attributed to the compositional variation between these two honeys.

Khalil *et al.* reported a DPPH inhibition of 54.64%–60.55% and a FRAP value of 348.33–445.00 µM·Fe(II)/kg for GH. In a later study, a radical scavenging activity of approximately 58% at a concentration of 60 mg/mL was reported for GH, along with a FRAP value of 325.79 µM·Fe(II)/100g [12]. The half maximum inhibitory concentration (IC_50_) of DPPH inhibition appeared to be 14.36 mg/mL with a FRAP value of 644.28 µM·Fe(II)/kg in another study [29]. Hussein *et al.* [15] revealed the FRAP values of 210.08–1108.90 µM·Fe(II)/kg in GH dissolved in distilled water and 188.97–1091.60 µM·Fe(II)/kg in GH dissolved in methanol. The DPPH free radical scavenging assay demonstrated GH dissolved in distilled water has an inhibition of 31.46%–82.68% as compared to the lower inhibition observed in methanolic GH (24.37%–79.26%). The study also reported that gamma irradiation increased total phenolic compounds and antioxidant activities of GH. Attending to the results from this study, methanol might not be suitable as a solvent for this honey. Methanol serves as an excellent solvent for polyphenols, however, the honey also contains other groups of bioactive constituents that are not well dissolved in methanol. In another study, GH exhibited an IC_50_ value of 6.68 mg/mL in a DPPH assay and a FRAP value of 115.61 µmol Fe(II)/100g, indicating that GH has a strong antioxidant capacity, ranked after Tualang honey [23].

Additionally, an AEAC (antioxidant equivalent ascorbic acid content) assay was used to estimate antioxidant content in GH and gave a result of about 317 mg/kg, which was far higher than that of Manuka honey [12]. Several studies reported the significant linear correlation between total phenolic content (TPC) and antioxidant capacities, with r values ranging between 0.789 and 0.965 for the DPPH assay and between 0.761 and 0.990 for the FRAP assay [12,13,22,29], confirming the contributions of phenolic compounds in antioxidant activities. However, other constituents also play important roles in combating free radicals. Ascorbic acid has a significant (*p* < 0.05) correlation with the DPPH assay (*r* = 0.542), whereas proline correlated significantly (*p* < 0.01) with the FRAP assay (*r* = 0.900) [12]. Significant correlations were also observed in protein with both DPPH and FRAP assays, with r values of 0.590 (*p* < 0.05) and 0.960 (*p* < 0.01), respectively [12].

Several groups of researchers have attempted to use animal models to confirm the putative anti-oxidative activities of GH. In 2011, Yao *et al.* [30] conducted a study to evaluate the anti-oxidative effects of GH in young and middle-aged rats. The results indicated that GH supplementation reduced DNA damage and plasma malondialdehyde (MDA). The decrease of MDA level was in a dose-dependent manner. GH supplementation modulated antioxidant enzyme activities in erythrocytes by increasing gluthathione peroxidase (GPx) activity and reducing catalase (CAT) activity in both young and middle-aged groups. The reduction of GPx and CAT activities were observed in the liver in both age groups, whilst SOD (superoxide dismutase) was decreased significantly (*p* < 0.05) in the young aged group only. Makpol *et al.* [31] also reported that pre-treatment with GH at 6 mg/ml reduced the DNA damage in gamma-irradiated human diploid fibroblasts (HDFs), but GH treatment given during and post-irradiation increased the level of DNA damage as compared to untreated control. Cell survival rate of HDFs decreased with increasing dose of gamma-ray exposure, and treatment with GH at pre- and during-radiation increased the cell survival rate. This indicates that GH acts as a protective agent in gamma-irradiated HDFs. Similarly, Sahhugi and coworkers [32] reported that GH supplementation reduced DNA damage and plasma MDA levels in young rats but not in aged rats when compared to their respective control group. GH supplementation also significantly increased cardiac superoxide dismutase activity in young rats and cardiac CAT activity in both young and aged rats. Meanwhile, there were no changes in erythrocyte SOD and GPx activity in both age groups as compared to their control counterparts, but CAT activity increased in young rats (*p* < 0.05).

Batumalaie *et al.* [33] used a different model to examine the antioxidant effects of GH. Hamster pancreatic cell lines (HIT-T15) were used and cultured in both normal and hyperglycemic conditions. Exposure of HIT-T15 cells to GH extract and its flavonoids (chrysin, luteolin, and quercetin) showed maximum cell viability at a concentration of 80 µg/mL and 80 µM, respectively. In hyperglycemic conditions, pretreatment of GH extract and its flavonoids increased cell viability significantly as compared to the cells cultured with glucose alone (*p* < 0.05). In a dose-dependent manner, the ROS generated in the presence of glucose was inhibited by GH extract and flavonoids. Pretreatment of GH extracts and flavonoids in individual cells under normal conditions has reduced the ROS activity in single cells in a dose-dependent manner. Lipid peroxidation was assayed using measurements of thiobarbituric acid-reactive substance (TBARS) and MDA. The results revealed that GH extract and its flavonoids reduced MDA production in the cells cultured in hyperglycemic circumstances. The F2 isoprostane production was measured to investigate the effect of GH extract and its flavonoids on the damage of free radicals induced by glucose in HIT-T15 cells. Unsurprisingly, pretreatment with GH extract and its flavonoids significantly reduced the production of F2 isoprostanes (*p* < 0.05) as compared to cells cultured in glucose alone. The researchers further examined the activities of GH and its bioactive constituents on insulin content. Pretreatment of GH honey and its flavonoids showed a significant (*p* < 0.05) elevation of insulin content as compared to the cells cultured in glucose alone.

## 5. Anti-Inflammatory Activities of GH

Inflammation is a complex physiological reaction of the body against infections, irritations, injuries, and cell damage, where it has vital roles in both innate and adaptive immunity [34,35]. Inflammation participates in a wide spectrum of chronic and degenerative diseases, such as rheumatoid arthritis, asthma, neurodegenerative diseases, inflammatory bowel disease, and cancer [36,37]. A series of pro-inflammatory cytokines are released during inflammation, namely interleukin 6 (IL-6), IL-12, tumor necrosis factor (TNF), interferon (IFN-ϒ), cyclooxygenase-2 (COX-2) and inducible nitric oxide synthase (iNOS), which initiate and amplify the inflammatory process [37,38,39]. Nuclear factor kappa B (NF-κB) is a transcription factor regulating the expression of various genes encoding pro-inflammatory mediators [40,41].

Compelling evidence from *in vivo* and *in vitro* studies have supported the anti-inflammatory property of GH. Kassim *et al*. [42] evaluated the inhibitory activity of GH on inflammatory mediators. GH and its extracts (methanol and ethyl acetate extracts) were used in the study. In a non-immune inflammatory and nociceptive model, rats’ paws were induced with carrageenan. Injection of lipopolysaccharide (LPS) was given to rats to study the anti-inflammatory effects of honey in an immune inflammatory model. Edema in the paw was measured using plethysmometer, and the results showed that GH and its extracts significantly reduced edema in both models (*p* < 0.05). The extracts (both methanol and ethyl acetate extracts) showed higher inhibition of edema in both models as compared with honey. This indicated that phenolic compounds play an important role in the inhibition of edema. The results showed that GH and its extracts also significantly reduced the pain indicated through the measurements of infrared withdrawal latency. The LPS groups had higher concentrations of nitric oxide (NO) and prostaglandin (PGE_2_) in exudates of paw tissues as compared with the carrageenan groups, exceptionally for the indomethacin groups which showed approximately the same quantity of inflammatory mediators. GH and its extracts significantly inhibited NO and PGE_2_ (*p* < 0.05).

Hussein *et al.* [43] conducted another study to examine the inhibition of GH against inflammation. Pretreatment with GH for 1 or 7 days, at a concentration of either 1 or 2 g/kg body weight, reduced the formation of paw edema significantly in a dose-dependent manner (*p* < 0.05). Plasma NO level increased in those inflammation-induced rats. However, oral feeding of GH at either dose significantly reduced the NO levels in rats with carrageenan-induced inflammation. The effect of 2 g/kg of body weight of honey administration was similar to that of the NSAID Indomethacin given at a dose of 10 mg/kg of body weight. In addition, GH significantly inhibited PGE_2_ production in both 1- and 7-day models. Plasma TNF-α and IL-6 were determined using ELISA, showing a decrease in the production of plasma TNF-α and IL-6. The effect of pretreatment at 2 g/kg body weight for 7 days was comparable to the effect of the NSAID Indomethacin. Real-time polymerase chain reaction (RT-PCR) and Western blot analysis were performed to evaluate the effect of GH on the expression of inflammatory-related enzymes (iNOS and COX-2), and pro-inflammatory cytokines (TNF-α and IL-6) genes and proteins in paw edema. GH at either dose significantly suppressed the gene expression of pro-inflammatory mediators (*p* < 0.05). Western blot analysis demonstrated that GH inhibited the protein expression of iNOS, COX-2, IL-6, and TNF-α in rat paw tissue in a dose-dependent manner.

The same group of authors has undertaken a different study to investigate the anti-inflammatory mechanism of GH in carrageenan-induced rat paw inflammation via the NF-κB pathway [44], the same doses (1 or 2 g/kg body weight) and duration (1 and 7 days) were used. The expression of NF-κB (p65 and p50) and inhibitors of κB (IκBα) genes in inflamed rat paws were significantly attenuated by pretreatment with GH in all doses and duration, and a greater down-regulating effect was observed in pretreatment with 2 g/kg of body weight of GH for 7 days—the effect was similar to that of Indomethacin. The cytoplasmic level of p65 and p50 increased significantly by the pre-treatment with GH, meanwhile the nuclear level of p65 and p50 decreased significantly with GH pre-treatment. Carrageenan injection induced the cytoplasmic degradation of the IκBβ protein. The cytoplasmic level of IκBβ significantly increased with pretreatment with GH. Histological analysis showed the infiltration of inflammatory cells was significantly decreased with treatment of GH. Immunohistochemistry assay revealed a significant reduction of TNF-α and COX-2 protein expression in the carrageenan-induced inflammation group.

To examine the anti-inflammatory effect of the intravenous injection of GH, Kassim *et al.* [45] performed a separate investigation using rats with LPS-induced endotoxemia. The authors reported a significant reduction in TNF-α levels after 4 h of GH treatment in rats injected with LPS. However, the effect was short-lived and was no longer observable at 24 h after treatment. It was noteworthy that honey showed strong inhibitory activity against IL-1β and IL-10 in the honey-treated group; there were significant differences in the levels of these two cytokines between the honey-treated groups and control group at 4 h and 24 h following treatment. Following GH treatment, IL-6 levels remained unchanged and similar to that of control group, while serum high motility group box 1 (HMGB1) protein levels decreased only at 24 h. A significant reduction in NO production was induced by GH at 4 h, and to a lesser extent at 24 h. Besides, GH was a potent inducer of hemeoxygenase-1 (HO-1), where significant differences between the honey-treated groups and the control group were evident at 4 h and at 24 h. The effort to explore the anti-inflammatory effects of GH was continued with a recent research work by Aziz *et al.* [46]. In a periodontitis-induced Sprague-Dawley rat model, GH was able to reduce 21.26% of plasma IL-1β and 81.27% of tissue IL-1β, suggesting the potential of GH in periodontal disease treatment.

An anti-inflammation mechanism of GH was proposed at the molecular level (Figure 1) [44]. GH might inhibit inflammation via suppression of the NF-κB pathway. The incoming signal pathway which activates the inhibitor of kappa β kinase (IKK) complex is blocked, disrupting phosphorylation, ubiquitination, and degradation of IκB proteins. This subsequently prevents the translocation of NF-κB dimers (p65 and p50) into the nucleus, and ultimately reduces the expression of inflammatory mediators such as iNOS, COX-2, TNF-α, and IL-6, as well as production of NO and PGE_2_.

## 6. Antimicrobial Properties of GH

Honey has been identified as a natural antimicrobial agent. This biological activity is attributed to its unique composition, containing both bee-origin glucose oxidase and non-peroxide constituents such as phenolic compounds, flavonoids, antibacterial peptides, methylglyoxal, methyl syringate, antibiotic-like derivatives, and other trace components [47,48,49]. Several studies have been conducted to screen the antibacterial properties of GH against bacterial strains.

In 2013, Zainol *et al.* [50] conducted an *in vitro* study to assay the antibacterial activity of GH along with few other Malaysian honeys. Comparatively, GH showed the lowest minimum inhibitory concentration (MIC) and minimum bactericidal concentration (MBC) values against all bacterial isolates in the broth dilution method. The MIC and MBC values for GH were 5%–15% (*w*/*v*) and 6.25%–15% (*w/v*), respectively. The agar well diffusion assay revealed that GH had a high antibacterial activity against *Bacillus cereus* with equivalent phenol concentration (EPC) measurements of 23.04% (*w*/*v*) in total activity and 22.31% (*w*/*v*) in non-peroxide activity. GH has also been reported to have excellent antibacterial activities in a recent study by Ng *et al.* [51]. GH was superior to Tualang and Durian honeys in terms of inhibitory activities against human pathogenic bacteria. GH exerted its inhibition to most of the tested strains from concentration of 40% undiluted honey. It was effective against *Klebsiella pneumonia, Staphylococcus aureus, S. epidermidis*, vancomycin-resistant enterococci (VRE) (*Enterococcus faecalis* and *E. faecium)*, *Escherichia coli*, and *Salmonella enterica* serovar Typhimurium. The MIC values were in the range of 125–1000 mg/mL, while MBC ranged 125–2000 mg/mL. The potency of the antibacterial properties of the honey was outlined as GH > Tualang honey > Durian honey. Meanwhile, the main author worked with other researchers to screen the effects of honey on *Enterococcus* spp. biofilm [52]. *Enterococcus* spp. biofilms formed on medical devices such as artificial hip prostheses, prosthetic heart valves, central venous catheters, intrauterine devices, and urinary catheters was a devastating medical problem [53]. Manuka honey was more effective in reducing established biofilm biomass as compared to GH. Despite this, GH was comparable to Manuka honey in terms of preventing biofilm formation of *Enterococcus* spp. GH has been reported to inhibit bacterial growth completely at 1 g/mL and exhibited the greatest reduction of biofilm biomass at a concentration of 0.5 g/mL. A comparative study was conducted to investigate the antibacterial activity of GH and Coconut honey against four selected bacterial species, namely *E. coli*, *S. aureus*, Methicillin-Resistant *S. aureus*, and Methicillin-sensitive *S. aureus* [54]. The growth of all bacterial strains was partially inhibited by the phenolic extract of GH at 1.3 mg/ml and completely inhibited at >1.95 mg/mL in a broth dilution assay. The disc diffusion assay revealed that GH extract inhibited all tested strains, starting from the lowest concentration, 0.65 mg, and the inhibition was in a dose-dependent manner. Both assays showed that GH had a more pronounced anti-bacterial activity than Coconut honey.

## 7. Anti-Proliferative and Chemo-Preventive Properties of GH

Cancerous cells lack of the ability to maintain a balance between apoptosis and cell renewal, in which the cells are continuously dividing, resulting in cellular and organ dysfunction. Researchers have attempted to discover novel natural products possessing anti-proliferative properties. A study conducted by Jubri *et al.* [55] aimed to study the anti-proliferative activity and apoptosis induction of GH on a liver cancer cell line (HepG2) and a normal liver cell line (WRL-68). MTS assay revealed the IC_50_ values of GH towards HepG2 and WRL-68 cells were 25% and 70%, respectively. In a bromodeoxyuridine (BrdU) assay, GH was found to reduce the proliferation of HepG2 at concentrations of 3% to 70%. The morphological analysis showed the reduction in cell size in GH-treated HepG2 cells. The findings substantiated the potential efficacy of GH as an anti-proliferative agent through apoptosis induction.

In a similar manner, Wen *et al.* [56] performed a study to observe the anti-proliferative effect of GH on the HT29 colon cancer cell line. Comparatively, the results indicated that GH was more potent than Nenas honey in suppressing the growth of colon cancer cells, with a lower IC_50_ of 39.0 mg/mL. The results showed that honey has a similar effect to the anti-inflammatory drug indomethacin in reducing PGE_2_ production by inflammation-induced cells. The findings suggested that GH is capable of suppressing the growth of HT29 colon cancer cells by inducing apoptosis and retarding inflammation. In another study [57], GH showed an anti-proliferative effect and induced cell death in all tested cancer cell lines (breast adenocarcinoma, MCF-7; human colorectal carcinoma, HCT-116; HepG2, and human alveolar basal epithelial adenocarcinoma, A549). Maximum cytotoxic effect was at 72 h on HCT116, with IC_50_ of 3.98% (*v*/*v*) followed by MCF-7 (IC_50_ = 6.31%).

In a very recent study, Tahir *et al.* [58] evaluated the synergistic effect of crude ginger extract and GH as potential chemo-preventive agents against HT29 cells. In the MTS assay, the IC_50_ of ginger and GH were 5.2 mg/mL and 80 mg/mL, respectively. The synergistic action of these two dietary agents was well depicted when the IC_50_ values of the combination treatment were lower for both ginger (3 mg/mL) and GH (27 mg/mL) with a combination index of <1. ELISA and RT-PCR were conducted, and the results revealed that cell death induced by the combined treatment was associated with the up-regulation of caspase 9 and IκBα gene expression, which stimulates early apoptosis. The expression of *KRAS*, *ERK*, *AKT*, *Bcl-xL*, and *NF-κB* (*p65*) genes was down-regulated by the combined treatment. The results suggested the combined treatment as a chemo-preventive agent to induce apoptosis of colon cancer cell. In another study [59], the chemo-preventive properties of the combined treatment were further explored using mTOR, Wnt/β-catenin, and apoptosis signaling pathways. The gene expressions of AKT, mTOR, Raptor, Rictor, β-catenin, GSK3β, TCF4, and cyclin D1 were down-regulated, whilst the gene expressions of cytochrome C and caspase 3 were up-regulated by the combined treatment. Attending to these results, the combination of GH and ginger might serve as a potential chemo-preventive agent by inhibiting mTOR and Wnt/β-catenin signaling pathways as well as the apoptosis induction pathway.

## 8. Wound Healing Properties of GH

Wound healing is a cascade of cellular and biochemical events, comprising three stages, namely inflammation, proliferation, and maturation [60]. Honey is one of the oldest and most enduring natural therapeutic agents in wound management, and can be administered topically or systematically and used alone or in combination with other substances [61,62]. The effort to explore the wound healing potential of GH was initiated by Aljadi *et al*. [60]. Biophysical and biochemical changes were observed in excision wounds of Sprague-Dawley rats. Quantitative biochemical analyses showed that a combination of topical and oral honey administration had a higher content of DNA, collagen, uronic acid, hexosamine, and serum albumin as compared to wounds treated with topical administration alone and control group. These measurements were consistent with the rate of wound contraction and epithelialization, in which the combined treatment group showed an accelerated wound healing. Rozaini and coworkers [63] evaluated tensile strength on burn wound healing treated with honeys in male Sprague-Dawley rats. Comparatively, the GH-treated group showed a higher value of tensile strength at day 3, 14, 21, and 28 days post injury as compared with the control group, but the value was lower than that of the Nenas honey-treated group. The same group of authors tested the wound healing effects of honeys using histological evaluation [64]. Inflammatory cells, especially neutrophils, were significantly lower, while proliferative cells (fibroblasts and endothelial) were significantly higher in GH-treated wounds as compared to control group (*p* < 0.05). These findings suggested that the honey accelerates dermal repair in wound healing. Another study [65] reported that topical application of GH significantly stimulated the rate of burn wound healing by increasing the rate of wound contraction. The effects were similar to Manuka honey-treated wounds.

Yusof and his colleagues [66] worked on the wound healing properties of GH and developed a honey hydrogel dressing for the enhancement of wound healing. The *in vivo* assessment showed that GH significantly stimulated wound contraction. Microscopic evaluations demonstrated a significant accelerated dermal repair in GH-treated wound, with an early attenuation of inflammatory reaction and reparative activities. A hydrogel with 6% honey was characterized by 337.04% elongation, tensile strength of 0.02647 mPa, pH of 4.3, and golden yellow color was formulated at the end of study as a wound dressing with healing properties. Efficacy of GH in wound healing was also evaluated by Tan *et al.* [67]. Intrasite gel containing modified carboxymethyl cellulose polymer, propylene glycol, and water was used as positive control, whereas saline was used as negative control. The results revealed that treatment with GH and Intrasite gel showed no significant difference in duration of wound healing; both healed in about 13 days as compared to around 16 days in the untreated group. Macroscopic examinations showed the changes in the GH-treated group from moist scab on day 5, detachment of scab in day 10, to fewer scars in day 15 of treatment. Meanwhile, the histological sections demonstrated new epidermis formed in GH-treated wounds was thinner and covered the entire wound area, providing protection to the wound from further injuries. The Intrasite gel-treated wounds had a faster epithelial regeneration, but the scab on the surface was thicker than that of GH-treated wounds. Wound contraction was also measured by the authors, indicating that GH increased the wound contraction, and the wound contractions were even greater than that of Intrasite gel-treated wound in day 10. The stimulation of GH in wound healing is achieved by providing energy for contractile activity, and enhancing the deposition of fibroblast and collagen, thus decreasing the scar deposition [60,68]. Fibroblasts play a vital role in tissue repair processes [69]. For this reason, a recent study was conducted by Aljadi *et al*. [70] to look into the effects of GH and its main constituents on the proliferation of cultured fibroblasts. The strongest effect of GH was noted at 6 h after treatment and a dose of 1.95 mg/mL, in which a 35% increase in cell viability over the control (*p* < 0.0001) was observed. The highest stimulatory effect of honey sugar was achieved at 132 mg/mL for 24 h with an increase of 16.6% in cell proliferation over control. Protein extract from honey had no direct effect on the growth of cultured fibroblasts, whilst preformed hydrogen peroxide exerted both stimulatory and inhibitory effects on the growth of cultured fibroblasts. Treatment with 0.57 µM preformed H_2_O_2_ for 2 h showed a significant increase in cell proliferation (11.5%) over the control (*p* < 0.05), whereas the greatest toxic effect was observed at 57 µM preformed H_2_O_2_ for 24 h. Similarly, continuous H_2_O_2_ generated by honey extract mixture (EM) constituted by sugar, protein extract, and phenolic extract could either stimulate or inhibit the growth of fibroblast *in vitro*. The 0.15 mg/mL treatment of EM showed the highest stimulatory effect on fibroblast proliferation at 2 h, where a 14.1% increase in cell growth was observed over the control. At 15 mg/mL of EM, the highest inhibitory effect was shown at 24 h with 22% of reduction in cell viability compared to that of control.

Rozaini *et al.* [71] observed the efficacy of honey hydrogel dressing with the incorporation of 6% GH on deep partial-thickness burns in terms of gross appearances, rate of wound contraction, and histological changes. The results demonstrated that wounds treated with honey hydrogel dressing had a better appearance and a significant enhancement in the rate of wound contraction (*p* < 0.05) as compared to the control group at 21 days after burn. Honey hydrogel also conferred a faster epithelialization in treated wounds. The authors continued to use the same formulation for further analysis on the wound healing effects of honey hydrogel using a molecular approach in another study [72]. RT-PCR revealed that the honey hydrogel treatment significantly (*p* < 0.05) suppressed the expression of pro-inflammatory mediators (IL-1α, IL-1β, and IL-6).

## 9. Other Health Benefits of GH

In an animal model, Samat *et al.* [73] observed the effects of GH acute administration on biochemical parameters. Rats fed with 2000 mg/kg body weight of GH did not show any adverse effects or deaths. The study reported a decrease in weight gain and energy efficiency, but a significant (*p* < 0.05) increase in total food intake and calories in female rats fed with GH when compared to control. In male rats fed with GH, a significant increase in body weight was observed. A significant (*p* < 0.05) decrease in triglycerides was displayed in both male and female rats fed with GH as compared to control. The results suggested the potential of GH in controlling weight gain and triglycerides. To study the effects of GH on the male reproductive system, Asiyah *et al.* [74] observed the changes of testis parameters and sperm quality in juvenile rats following a 60-day GH supplementation (1.0 mL/100 g body weight daily). The results showed that the GH-treated group has significantly (*p* < 0.05) higher sperm motility (18.85 × 10^5^/mL) and number of normal sperm morphology (193.73) than control group. Meanwhile, the testicular parameters (weight, length, and width) showed no significant changes.

## 10. Conclusions

Being a natural food, honey is highly appreciated in different cultures. The characterization of the health benefits of honey has confirmed the potential of this natural food as medicine. From the evidence presented in this paper, GH has apparently demonstrated a wide spectrum of biological properties, suggesting its feasibility in clinical applications. However, the clinical intervention studies are still lacking at this stage. A great deal of scientific effort is expected in GH research in order to provide some convincing evidence to support the sustainable use of honey through human trials. Determination of appropriate dosage of GH should be prompted to avoid undesirable side effects. The mechanisms of honey and its active constituents on disease management are still inconclusive; urgency to call for in-depth research ought to be an immediate action. Since phenolic compounds are one of the most interesting groups of chemical compounds studied in honey, future studies should also emphasize the role of this group of active constituents in biological activities. Chemometric methods are required to characterize a complex profile of polyphenols in GH, and isolated compounds should be tested for their possible health benefits.

## Figures and Tables

**Figure 1 medsci-04-00003-f001:**
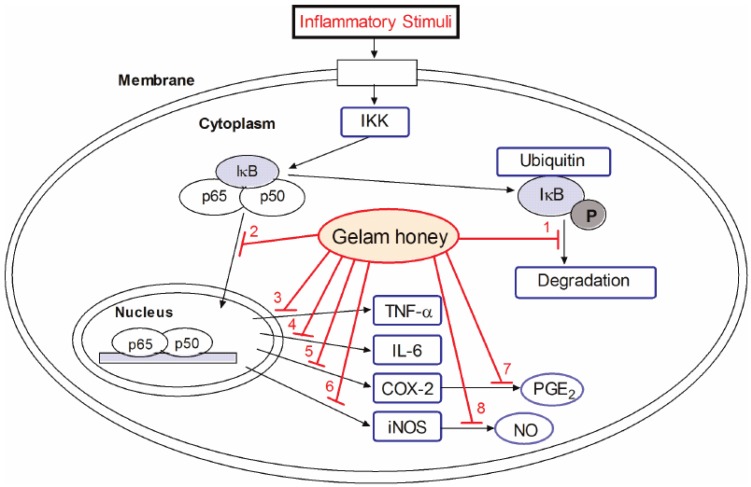
Proposed mechanism by which GH exerts inhibition on inflammation. GH inhibits (1) the degradation of inhibitors of κB (IκBα) that causes (2) nuclear translocation of Nuclear Factor κB (NF-κB) dimer (p50 and p65), resulting in the reduced expression of inflammatory mediators, including (3) Tumor Necrosis Factor alpha (TNF-α); (4) Interleukin 6 (IL-6); (5) Cyclooxygenase 2 (COX-2); and (6) inducible nitric oxide synthase (iNOS). The down-regulation of COX-2 and iNOS expressions decrease the productions of (7) Prostaglandin (PGE_2_) and (8) Nitric Oxide (NO) [44].

**Table 1 medsci-04-00003-t001:** Physiochemical characteristics of Gelam honey (GH).

Physicochemical Characteristics	Values
pH	3.55–3.91
Free acids (meq/kg)	32.33–50.93
Lactones (meq/kg)	5.34–9.00
Moisture content (%)	17.93–20.76
Electrical conductivity (mS/cm)	0.74
Total dissolved solids (ppm)	368.33
Color intensity/ ABS_450_ (mAU)	500.30–1355.00
Color characteristic (mm Pfund)	122.00, Dark amber
HMF content (mg/kg)	8.52–66.00

* HMF: Hydroxymethylfurfural.

**Table 2 medsci-04-00003-t002:** Chemical composition of GH.

Chemical Compound	Values	Reference
**Carbohydrate**		[12,15]
Total sugar content (%)	64.93–69.60	
Reducing sugar (%)	62.17–69.16	
Sucrose (%)	0.41–2.77	
**Protein/Amino Acid Content**		[12]
Protein content (g/kg)	3.14	
Proline content (mg/kg)	261.33	
**Mineral Content (mg/kg)**		[15,19]
Sodium	17.37–196.84	
Potassium	23.04–1363.40	
Calcium	21.63–275.77	
Iron	2.37–142.37	
Magnesium	4.94–31.63	
Zinc	4.91–29.23	
Copper	0.29–2.21	
Selenium	16.20	
**Vitamin Content (mg/kg)**		[15,20,21]
Thiamin	13.85	
Riboflavin	94.21	
Nicotinic acid	355.38	
Panthotenic acid	12.93	
Ascorbic acid	22.90–67.36	
Vitamin E (µg/g)	55.59–70.70	
**Polyphenol Content**		[12,13,20,22,23]
Total phenolic content (TPC)	34.30–159.74 mg GAE/100 g; 8.47–71.51 mg RE/100 g	
Total flavonoid content (TFC)	1.47–32.89 mg RE/100g; 3.24–4.30 mg CE/100g; 46.11mg QE /100 g	
**Phenolic Compounds (µg/100 g)**		[22]
Gallic acid	859.43–876.80	
Chlorogenic acid	502.77–528.08	
Caffeic acid	428.84–442.01	
p-coumaric acid	301.45–308.31	
Ferulic acid	356.93–381.37	
Ellagic acid	558.78–575.67	
Quercetin	1588.90–1594.30	
Hesperetin	1475.20–1477.78	
Chrysin	1498.60–1504.60	
**Enzymes**		[21]
Invertase (U/L) ^†^	85.56	
Diastase (DN) *	0.57	

* One DN is expressed as a diastase unit per gram of honey; ^†^ One unit of invertase activity is equivalent to the formation of 1 µmol glucose per minute at pH 4.5; TPC is expressed as mg gallic acid equivalent (GAE) or mg rutin equivalent (RE)/100 g; TFC is expressed as RE or catechin equivalent (CE) or quercetin equivalent (QE)/100 g.

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
