# Peer review of "Insights into Putative Health Implications of Gelam (Melaleuca cajuputi) Honey: Evidence from In-Vivo and In-Vitro Studies"

_medsci, 2016, doi:10.3390/medsci4010003_

Round 1

Reviewer 1 Report

The manuscript is a work of revision, mainly focusing in the biological activities of the Gelam honey. Although it is an interesting manuscript, the theme is mainly directed to those who work with Gelam honey. My conclusion is that the work should be completed with more chemical information, for instance, with the profile of phenolic compounds, since it is the most studied class of compounds in honey.

Author Response

Reviewer 3:

We appreciate on the suggested conclusion, and we have incorporated this valuable cinformations in this manuscript in order to give a better recommendations for future scientific efforts that are pertinent to Gelam honey and even other honeys.

Reviewer 2 Report

It is a very well read thorough review of the composition and potential of Gelam Honey

Author Response

There was no particular comments to be amended from reviewer 2

Reviewer 3 Report

The authors write the review of “ Novel insights into putative health implication of gelam (Melaleuca cajuputi) Honey: Evidence from In-vivo and In-vitro studies. The manuscript is interesting and gives information on the Gelam Honey (GH). Their article had given a brief explanation on GH honey attributes to familiarize reader. However, the use of the English language will still require considerable attention.

For the benefit of the reader, however, a number of points need justification. These are given below.

1. In the title of this manuscript, the authors use the “Novel Insights into…” this manuscript is review paper and all data are been published, since it is not suitable use the “Novel” this word. Please rewrite it!     

2. In the title “(Melaleucacajuputi)” should revise to “(Melaleuca cajuputi)”,

Please check in this manuscript,  some typing errors have to be corrected.

The quality of this manuscript is not solid enough for publishing in the Journal.

3.  In the introduction, line 9, “ Apisdorsatabees”  should revised to “ Apis dorsata” ; line 10 “(Melaleucacajuputi)” should revise to “(Melaleuca cajuputi)” .

4. In the page 6, line 3 to line 7, those data were describe or from Gelam Honey, it should clear write it’s from GH or for GH.

Author Response

Reviewer 1:

1.      We have amended the title and removed the word “novel” as suggested.

2.      We are sorry for the typing errors of all the scientific names in this manuscript. We have corrected them accordingly.

3.      For those data describing GH in page 6, we have written it clearly in order to avoid confusion.